# Scale-Space Hypernetworks for Efficient Biomedical Imaging

**Jose Javier Gonzalez Ortiz**[*]
MIT CSAIL
Cambridge, MA
josejg@mit.edu

**John Guttag**
MIT CSAIL
Cambridge, MA
guttag@mit.edu

**Adrian V. Dalca**
MIT CSAIL & MGH, HMS
Cambridge, MA
adalca@mit.edu

## Abstract

Convolutional Neural Networks (CNNs) are the predominant model used for a variety of medical image analysis tasks. At inference time, these models are computationally intensive, especially with volumetric data. In principle, it is possible to trade accuracy for computational efficiency by manipulating the rescaling factor in the downsample and upsample layers of CNN architectures. However, properly exploring the accuracy-efficiency trade-off is prohibitively expensive with existing models. To address this, we introduce Scale-Space HyperNetworks (SSHN), a method that learns a spectrum of CNNs with varying internal rescaling factors. A single SSHN characterizes an entire Pareto accuracy-efficiency curve of models that match, and occasionally surpass, the outcomes of training many separate networks with fixed rescaling factors. We demonstrate the proposed approach in several medical image analysis applications, comparing SSHN against strategies with both fixed and dynamic rescaling factors. We find that SSHN consistently provides a better accuracy-efficiency trade-off at a fraction of the training cost. Trained SSHNs enable the user to quickly choose a rescaling factor that appropriately balances accuracy and computational efficiency for their particular needs at inference.

## 1 Introduction

Convolutional Neural Networks (CNNs) are a widely-used and fundamental tool in medical image analysis [2, 29, 37, 52, 58]. However, their practical deployment is hindered by their high computational requirements, especially in resource-constrained environments often encountered in medical applications [65, 71]. This has led to techniques to lower computational demands while preserving model accuracy. Popular strategies to address this issue encompass quantization [28, 33, 56], pruning [5, 19, 22, 35], and factoring [10, 34, 70], which focus on reducing the parameter count and the size of the convolutional kernels. These techniques invariably introduce a trade-off between model accuracy and computational efficiency. To understand this trade-off, developers need to train many models and select the best performing one based on a balance of the aforementioned metrics.

In this work, we propose an alternative approach aimed at improving the computational efficiency of prevalent medical imaging models, such as U-Net networks, which internally resize image features at different scales [58]. While existing convolutional architectures usually reduce the spatial scale by a fixed factor of two [24, 27, 40, 66], we study more general architectures that enable this factor to vary continuously. By reducing the spatial dimensions of intermediate features, we can reduce the computational cost at *both training and inference* stages, as the convolutional cost is directly linked to the size of the intermediate spatial feature maps.

---

[*]Corresponding author

37th Conference on Neural Information Processing Systems (NeurIPS 2023).

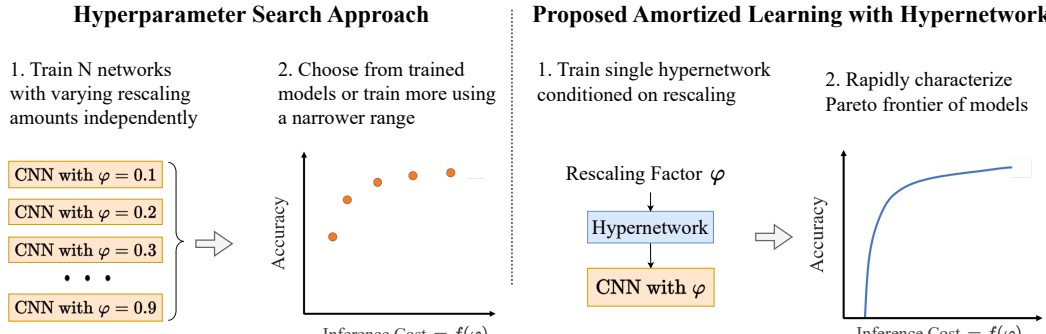

Figure 1: Strategies for exploring the accuracy-efficiency trade-off of varying the CNN rescaling factor. A standard approach (*left*) requires training N models with different rescaling factors independently. Our proposed amortized learning strategy (*right*) exploits the similarity between the tasks and trains a single hypernetwork model. Once trained, we can rapidly evaluate many rescaling settings and efficiently characterize the Pareto frontier between model accuracy and computational cost. This enables rapid and precise model choices depending on the desired trade-off at inference.

However, like with other techniques for reducing computation, characterizing the trade-off between accuracy and efficiency for a range of rescaling factors can be computationally demanding, as it requires training multiple independent models. To address this challenge, we introduce a hypernetwork learning framework, Scale-Space Hypernetworks (SSHN), which enables a *single* model to capture a complete landscape of models representing a continuous range of rescaling ratios. Once trained, this single SSHN model can be used to rapidly generate the Pareto frontier for the trade-off between accuracy and efficiency as a function of the rescaling factor (Figure 1). This enables a user to choose, depending on the inference setting, a rescaling factor that appropriately balances accuracy and computational efficiency for their use case.

We evaluate our approach using several medical image segmentation and registration tasks including different biomedical domains and image types. We demonstrate that the proposed hypernetwork based approach is able to learn models for a wide range of feature rescaling factors, and that inference networks derived from the hypernetwork perform at least as well, and in most cases better than, networks trained with fixed rescaling factors. SSHNs enable us to find that a wide range of rescaling factors achieve similar accuracy results despite having substantially different inference costs.

Our main contributions are:

- We introduce Scale-Space HyperNetworks (SSHN), a single model that, given a rescaling factor, predicts the weights for a segmentation network that uses that amount of rescaling in the spatial downsampling and upsampling layers. By reducing the spatial dimensions of intermediate features, we reduce the inference cost.

- We evaluate our method on several medical image analysis tasks and show that this approach makes it possible to characterize the trade-off between model accuracy and inference efficiency far faster and more completely than existing approaches.

- Using SSHNs, we demonstrate that on a variety of medical imaging tasks, many rescaling factors lead to similar results despite having substantially different inference costs. We use SSHN to rapidly explore this trade-off for a dataset of interest, showing that it can lead to 50% inference FLOPs reduction without sacrificing model accuracy.

- We show that learning a single model with varying rescaling factors has a regularization effect, which consistently improves the accuracy of networks trained using our method with respect to networks trained with a fixed rescaling factor. For some tasks, SSHN consistently outperforms regular models by 2-3% for the same rescaling factor.

## 2  Related Work

**Resizing in Convolutional Neural Networks**. Resizing layers are a fundamental operation common to most modern convolutional neural architectures [24, 55, 66, 78]. These are used to downscale or upscale the spatial dimensions of intermediate feature maps, providing a way for the network to aggregate visual context at multiple scales. Resizing layers have been implemented in a variety of ways, including max pooling, average pooling, bilinear sampling, and strided or atrous convolutions [3, 8, 9, 63]. Recent work has investigated using a wider range of rescaling operations by stochastically downsampling intermediate features [20, 42] or learning differentiable resizing modules that replace resizing operations in the network [36, 45, 57]. These techniques learn the differentiable modules as part of the training process, optimizing them to improve model accuracy. The final result is a model with features optimally resized for the data available at training. In contrast, our goal is to learn a landscape of models with varying rescaling factors that characterize the entire trade-off curve between accuracy and efficiency. This also enables users to choose an inference-time resizing factor that appropriately balances efficiency and accuracy for their dataset.

**Hypernetworks**, neural network models that generate the weights for another neural network [21, 39, 60], are effective models that have been successfully used in a wide range of applications. These include neural architecture search [6, 72], Bayesian optimization [41, 53], weight pruning [46], continual learning [68], multi-task learning [61] meta-learning [73] and knowledge editing [11]. Recently, hypernetworks have been employed in hyperparameter optimization problems [25, 47, 49, 69]. In these gradient-based optimization approaches, a hypernetwork is trained to predict the weights of a primary network conditioned on a given hyperparameter value. Our work shares similarities with hypernetwork-based neural architecture search techniques, as both strategies explore a multitude of potential architectures during training [17]. However, our work features a key distinction: we do not *optimize* the architectural properties of the network during training. Instead, we focus on learning a variety of architectures jointly, and enable choosing the architecture during inference based on the needs of the user.

## 3  Scale-Space HyperNetworks

Convolutional Neural Networks (CNNs) are parametric functions $\hat{y} = f(x; \theta)$ that map input values $x$ to output predictions $\hat{y}$ using a set of learnable parameters $\theta$. The network parameters $\theta$ are optimized to minimize a loss function $\mathcal{L}$ given dataset $\mathcal{D}$ using stochastic gradient descent strategies. For example, in supervised scenarios,

$$\theta^* = \arg\min_{\theta} \sum_{x,y \in \mathcal{D}} \mathcal{L}\left(f(x; \theta), y\right). \tag{1}$$

Most CNNs architectures use some form of rescaling layers with a *fixed* rescaling factor that change the spatial size of intermediate network features, most often by halving (or doubling) along each spatial dimension. We define a generalized version of convolutional models $f_\varphi(x, \theta)$ that perform feature rescaling by a *continuous* factor $\varphi \in [0, 1]$. Specifically, $f_\varphi(x; \theta)$ has the same parameters and operations as $f(x; \theta)$, with all intermediate feature rescaling operations being determined by the factor $\varphi$. For instance, in existing image models, downsampling layers $R(x)$ map an input tensor of shape $(C, H, W)$ to an output tensor of shape $(C, H/2, W/2)$ where $H$ and $W$ are the height and width of the feature map and $C$ is the channel dimension. In contrast, we consider resizing layers of the form $R_\varphi(x)$ that produce an output tensor of shape $(C, \lceil \varphi H \rceil, \lceil \varphi H \rceil)$.

To characterize model accuracy as a function of the rescaling factor $\varphi$ using standard techniques requires training multiple instances of $f_\varphi(\cdot; \theta)$, one for each rescaling factor, leading to substantial computational cost. Instead, we propose a framework to learn a family of parametric functions $f_\varphi(\cdot; \theta)$ jointly, using an amortized learning strategy that learns the effect of rescaling factors $\varphi$ on the network parameters $\theta$. We employ a function $h(\varphi; \omega)$ with learnable parameters $\omega$ that maps the rescaling ratio $\varphi$ to a set of convolutional weights $\theta$ for the function $f_\varphi(\cdot; \theta)$. We model $h(\varphi; \omega)$ with another neural network, or *hypernetwork*, and optimize its parameters $\omega$ based on the learning objective

$$\omega^* = \arg\min_{\omega} \sum_{x,y \in \mathcal{D}} \mathcal{L}\left(f_\varphi(x; h(\varphi; \omega)), y\right). \tag{2}$$

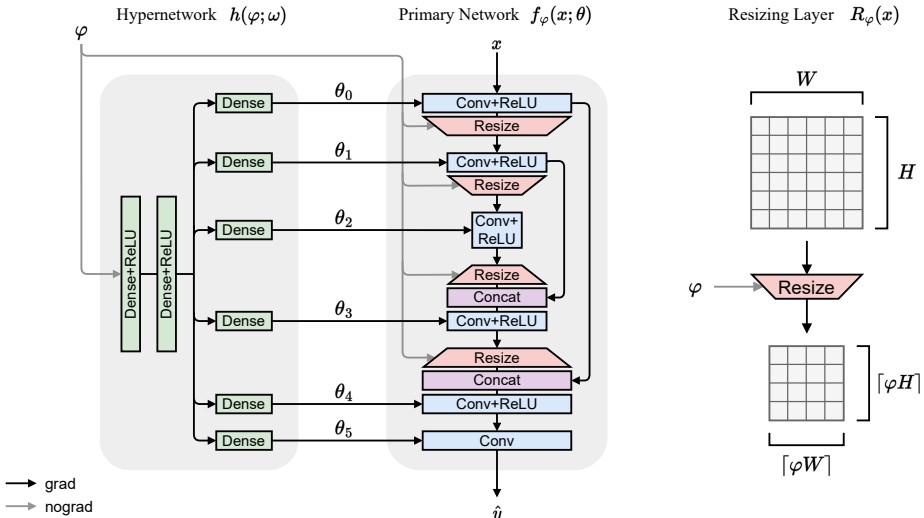

Figure 2: Diagram of the proposed hypernetwork-based model used to jointly learn a family of segmentation networks with flexible feature rescaling. At each iteration, a rescaling factor $\varphi$ is sampled from a prior distribution $p(\varphi)$. Given $\varphi$, the hypernetwork $h(\varphi; \omega)$ predicts the parameter values $\theta$ of the primary network $f_\varphi(x; \theta)$. Given a sample datapoint $x$, we predict $\hat{y} = f_\varphi(x; \theta)$ and compute the loss $\mathcal{L}(\hat{y}, y)$ with respect to the true label $y$. The resulting loss gradients only lead to updates for learnable hypernetwork weights $\omega$. The hypernetwork is implemented as a Multi Layer Perceptron (MLP) and the primary network uses a UNet-like architecture. On the right, we illustrate how the resizing layer $R_\varphi(x)$ variably resizes the spatial dimensions of a given input based on the scale factor $\varphi$.

At each training iteration, we sample a resizing ratio $\varphi \sim p(\varphi)$ from a prior distribution $p(\varphi)$, which determines both the rescaling of internal representations and, via the hypernetwork, the weights of the primary network $f_\varphi(\cdot; \theta)$.

The hypernetwork model has more trainable parameters than a regular primary network, but the number of convolutional parameters $\theta$ is the same. At inference time, we leave out the hypernetwork entirely, using the predicted weights for a given $\varphi$, incurring only the computational requirements of the primary network.

**Implementation**. We model the hypernetwork $h(\varphi; \omega)$ using a fully connected neural network with several layers, a common choice in hypernetwork literature [7, 21, 25, 67, 68]. We encode the hypernetwork input as a vector $[\varphi, 1 - \varphi]$ to prevent biasing the fully connected layers towards the magnitude of $\varphi$. Given an input value $\varphi$, the hypernetwork $h$ predicts a set of outputs $\{\theta_0, \theta_1, \ldots, \theta_K\}$ corresponding to the convolutional filters of the primary network. Each weight is predicted as a flat vector and then reshaped to the appropriate dimensions in the primary network. In the applications in our experiments, the primary network follows a UNet-like structure, the most prevalent and widely-used design in medical imaging tasks [30, 58]. We modify the architecture by replacing the discrete downsampling and upsampling layers with variable resizing layers determined by $\varphi$, using differentiable bilinear interpolation operations. Figure 2 illustrates Scale-Space HyperNetworks applied to a three stage U-Net architecture.

## 4 Experimental Setup

### 4.1 Tasks

We evaluate on two fundamental medical image analysis tasks: segmentation and registration.

**Segmentation**. In segmentation tasks, the primary network $f(x; \theta) \to y$ maps an input scan $x$ to an output segmentation map $y$. We use supervised training and optimize an objective of the form $\mathcal{L}(\hat{y}, y)$, where $y$ is the ground truth segmentation map and $\hat{y}$ is the predicted segmentation map. In our

experiments, we first pre-train the networks using a categorical cross-entropy loss and then finetune them using a soft Dice-score loss [51, 64]. We evaluate each segmentation method using the Dice score [12], which quantifies the overlap between two regions and is widely used in the segmentation literature. Dice is expressed as $\text{Dice}(y, \hat{y}) = (2|y \cap \hat{y}|)/(|y| + |\hat{y}|)$. A Dice score of 1 indicates perfectly overlapping regions, while 0 indicates no overlap. For datasets with more than one label, we average the Dice score across all foreground labels.

**Registration**. In registration tasks, a moving image $x_m$ is registered to a fixed image $x_f$ by applying a flow or deformation field $\phi$, like $\hat{x}_f = x_m \circ \phi$. Existing work has shown that a parametric model $f(x_m, x_f; \theta) \to \phi$ can be optimized to learn this mapping [2]. This is commonly achieved with an unsupervised objective that optimizes both an image alignment term $\mathcal{L}_{\text{sim}}$ between the fixed and *moved* image $x_m \circ \phi$, as well as a term encouraging spatial regularization of the flow field: $\mathcal{L}_{\text{reg}}$. The learning objective is then $\mathcal{L} = \mathcal{L}_{\text{sim}}(x_m \circ \phi, x_f) + \lambda_{\text{reg}}\mathcal{L}_{\text{reg}}(\phi)$, where $\lambda_{\text{reg}}$ controls the amount of flow-field regularization. We follow the experimental setup as in [2], using a U-Net architecture for the primary (registration) network, MSE for $\mathcal{L}_{\text{sim}}$ and total variation for $\mathcal{L}_{\text{reg}}$. For evaluation, we use the predicted flow field to warp anatomical segmentation label maps of the moving image, and measure the volume overlap to the fixed label maps.

### 4.2 Datasets

We use four popular and quite different biomedical imaging datasets OASIS [25, 50], PanDental [1], WBC [76] and CAMUS [43]. **OASIS** consists of 414 brain MRI scans that have been skull-stripped, bias-corrected, and resampled into an affinely aligned, common template space. For each scan, segmentation labels for 24 brain substructures from the FreeSurfer protocol [18] in a 2D mid-coronal slice are available. **PanDental** includes 215 XRay scans of the lower half of patients' heads, with a label for the entire mandible and a label for all teeth. **CAMUS** is an ultrasound dataset, containing 2D apical four-chamber and two-chamber view sequences acquired from 500 patients. **WBC** consists of 400 microscopy images of white blood cells, with labels for the cell nucleus and cytoplasm. In each dataset, we trained using a (64%, 16%, 20%) train, validation, test split of the data.

### 4.3 Baseline Methods

We compare our method against three other strategies for creating an efficiency-accuracy Pareto curve: Fixed, Stochastic, and FiLM.

**Fixed**. We train a set of conventional U-Net models with fixed rescaling factors. These are identical to the primary network, and are trained using a fixed rescaling factor $\varphi$ (i.e., no hypernetwork is used). We train several baseline models at 0.05 rescaling factor increments from 0 to 0.5.

**Stochastic**. In recent work, convolutional networks were trained with a stochastic rescaling factor, leading to improved generalization [42]. We incorporate a standard U-Net baseline, where scale factor $\varphi$ is stochastically sampled during training. Under this setting, a single model is learned, whose parameters $\theta$ are optimized to work with all possible rescaling factors within the sampled range.

**FiLM** Feature-wise Linear Modulation (FiLM) layers are a conditioning mechanism for convolutional models [16, 54]. FiLM modules map an input $z$ to a series of multiplicative and additive vectors $\gamma$ and $\beta$ that are used to affinely transform the intermediate feature maps at different points of the network. Recent work has used this mechanism to learn models in a amortized way, by stochastically sampling the conditioning input $z$ during training and thus enabling a Pareto frontier of models [13]. We test a model incorporating FiLM operations after every convolutional operation. Similar to our approach, the scale factor $\varphi$ is stochastically sampled during training, and used to modulate the FiLM modules in the primary network.

### 4.4 Experimental Details

**Primary Network Architecure**. For our experiments, we use the U-Net architecture, since it is the most widely-used segmentation CNN architecture in the medical imaging literature [30, 58]. We use bilinear interpolation layers to downsample or upsample a variable amount based on the scale factor $\varphi$ for downsampling. For the downsampling layers we multiply the input dimensions by $\varphi$ and round to the nearest integer. For the upsampling layers we upsample to match the size of the feature map that is going to be concatenated with the skip connection. We use five encoder stages and four

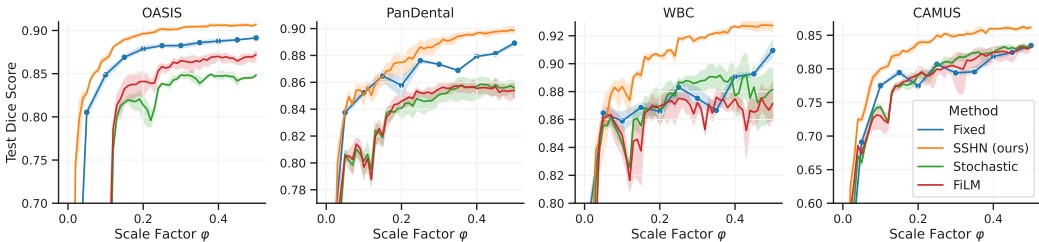

Figure 3: **Accuracy Results**. Comparison between a family of networks trained with fixed amounts of feature rescaling (Fixed), our proposed SSHN method and the other variable resizing methods (Stochastic, FiLM). We report results in the test set for the four considered segmentation datasets. In all settings the SSHN model outperforms the individually trained models (Fixed), whereas the other dynamic resizing baseline fail to meaningfully improve upon them.

decoder stages, with two convolutional operations per stage and LeakyReLU activations [48]. For networks trained on OASIS Brains we use 32 features at each convolutional layer. We found that the results we discuss below held across various architectural settings. **Hypernetwork**. We implement the hypernetwork using three fully connected layers. Given a vector of $[\varphi, 1 - \varphi]$ as input, the output size is set to the number of convolutional filters and biases in the segmentation network. We use the vector representation $[\varphi, 1 - \varphi]$ to prevent biasing the fully connected layers towards the magnitude of $\varphi$. The hidden layers have 10 and 100 neurons respectively and LeakyReLU activations [48] on all but the last layer, which has linear outputs. Hypernetwork weights are initialized using Kaiming initialization [23], and bias vectors are initialized to zero. **Evaluation**. For each experimental setting we train five replicas with different random seeds and report mean and standard deviation. Once trained, we use the hypernetwork to rapidly evaluate a range of $\varphi$, using 0.01 intervals. This is more fine-grained than is feasible for the Fixed baselines, since all evaluations for our method use the same model, whereas the Fixed baseline requires training separate models for each rescaling factor. **Training**. We use the Adam optimizer [38] and we train networks until the loss in the validation set stops improving. During training we sample the rescaling from $\mathcal{U}(0, 0.5)$ We provide additional platform and reproducibility details in the supplement, and we will publicly release our code.

## 5    Experimental Results

We present two sets of experiments. The first set evaluates the accuracy and efficiency of segmentation models generated by our hypernetwork, both in absolute terms and relative to models generated with fixed resizing. In the second set of experiments we study why SSHN achieve improved results.

### 5.1    Accuracy and Computational Cost

**Accuracy**. First, we assess the ability of the hypernetwork model to learn a continuum of primary network weights for various rescaling factors, and evaluate how the performance of models using those weights compare to that of models trained independently with specific rescaling factors. Figure 3 shows the Dice score for the segmentation task, on the held-out test split of each dataset, as a function of the rescaling factor $\varphi$. We see that in some cases many rescaling factors achieve similar performance. For example, in the OASIS task, SSHN achieves a mean Dice score across brain structures above 0.89 for most rescaling factors–suggesting that inference cost can be reduced without sacrificing accuracy.

The Stochastic and FiLM models fail to consistently match the performance of the Fixed baselines. In contrast, the networks predicted by SSHN are consistently more accurate than equivalent networks trained with specific rescaling factors, suggesting that the SSHN models, unlike the other approaches, are successfully adapting the predicted weights based on the scaling factor. Table 5 shows the best performing model variant found for each approach.

Figure 6 presents results for registration of the OASIS dataset. Again, SSHN networks out perform the independently trained networks. In contrast, the Stochastic and FiLM methods perform slightly worse than the Fixed baseline. Experiments later in the paper delve deeper into this improvement in model generalization.

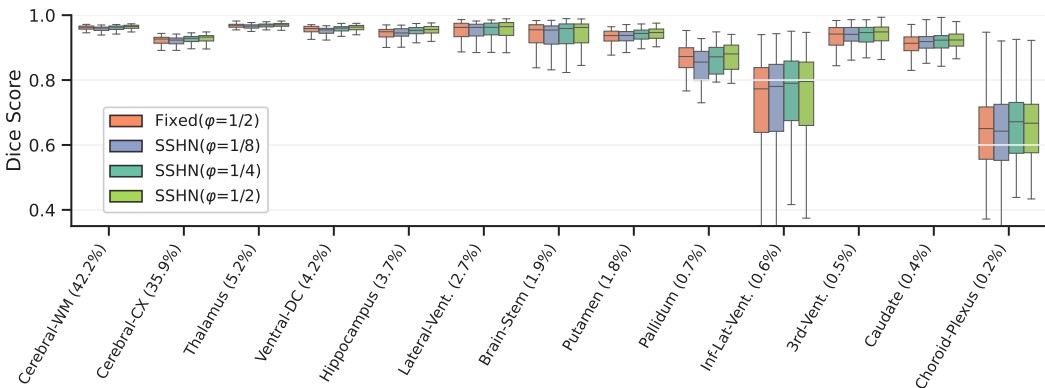

Figure 4: Dice coefficient across various brain structures for the baseline method (Fixed) trained and evaluated at $\varphi = 0.5$, and the SSHN approach evaluated at various rescaling factors $\varphi$ for the OASIS segmentation task. Anatomical structures are sorted by mean volume in the dataset (in parentheses). Labels consisting of left and right structures (e.g. Hippocampus) are combined. We abbreviate the labels: white matter (WM), cortex (CX) and ventricle (Vent). The SSHN model achieves similar performance for all labels despite the reduction in feature map spatial dimensions.

We also explore how varying the rescaling factor $\varphi$ affects the segmentation of smaller labels in the image using the OASIS dataset. Figure 4 presents a breakdown by neuroanatomical structures for a Fixed baseline with $\varphi = 0.5$ and a subset of rescaling factors for the SSHN model. The proposed hypernetwork model achieves similar performance for all labels regardless of their relative size, even when downsampling by substantially larger amounts than the baseline.

**Computational Cost**. We analyze how each rescaling factor affects the inference computational requirements of each segmentation network. We measure the number of floating point operations (FLOPs) required to use the network for each rescaling factor at inference time. We also evaluate the amount of time used to train each model.

Figure 7 depicts the trade-off between test Dice score and the inference computational cost for all segmentation tasks, and Figure 6 shown the results in the registration task on OASIS. We observe that the choice of rescaling factor has a substantial effect on the computational requirements. For instance, for the OASIS and PanDental tasks, reducing the rescaling factor from the default $\varphi = 0.5$ to $\varphi = 0.4$ reduces computational cost by over 50% and by 70% for $\varphi = 0.3$. In all cases, we again find that hypernetwork achieves a better accuracy-FLOPs trade-off curve than the set of baselines. Table 1 in the supplement report detailed measurements for accuracy (in Dice score), inference cost (in inference GFLOPs) and training time (in hours) for all methods approaches.

Table 5: **Segmentation Results**. Test Dice score values for the different methods evaluated on the considered segmentation tasks. Standard Deviation is reported across training replicas.

| Method | CAMUS | OASIS | PanDental | WBC |
|---|---|---|---|---|
| Fixed | $83.5 \pm 0.3$ | $89.1 \pm 0.0$ | $88.9 \pm 0.2$ | $90.9 \pm 0.6$ |
| Stochastic | $83.3 \pm 0.1$ | $84.8 \pm 0.2$ | $85.6 \pm 0.7$ | $88.1 \pm 2.0$ |
| FiLM | $83.2 \pm 0.3$ | $87.2 \pm 0.6$ | $85.4 \pm 0.6$ | $87.1 \pm 1.4$ |
| SSHN (ours) | $\mathbf{86.2 \pm 0.3}$ | $\mathbf{90.7 \pm 0.1}$ | $\mathbf{89.9 \pm 0.3}$ | $\mathbf{92.7 \pm 0.4}$ |

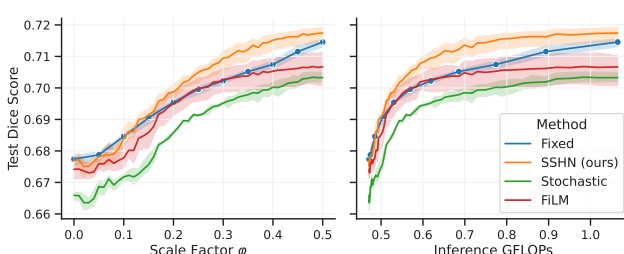

Figure 6: **Registration Results**. Results for the learnable registration task on OASIS. Test set results as a function of the rescaling factor (*left*) and as a function of the computation cost, in GFLOPs (*right*) for the considered methods. The SSHN model matches the independently trained networks (Fixed) for most values of the rescaling factor $\varphi$.

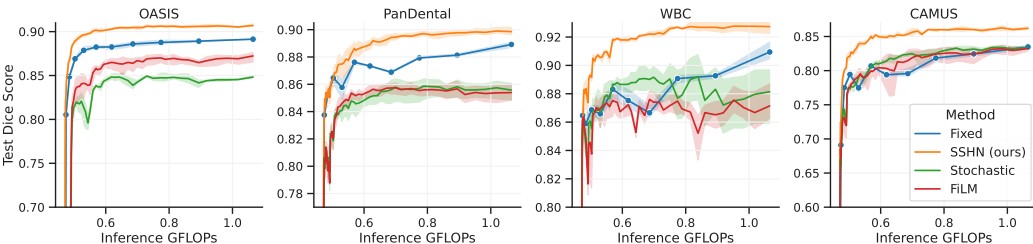

Figure 7: **Efficiency Results**. Trade-off curves between test model accuracy (Dice score) and inference computational cost (GFLOPs). SSHN models successfully characterize the trade-off between model accuracy and efficiency while outperforming models trained with fixed amounts of feature rescaling (Fixed). In all datasets, the inference computational cost of SSHN can be reduced by at least 40% without a significant change in model accuracy. Results are averaged across network initializations, and variance is indicated with shaded regions.

Because of identical inference cost (for a given rescaling factor) and improved segmentation quality of SSHN, its Pareto curve dominates the networks with fixed resizing. SSHN therefore requires less inference computation for comparable model quality. For example, for segmentation networks trained on OASIS, for $\varphi \in [0.25, 0.5]$ there is no loss in quality, showing that a more than 20% reduction in FLOPS can be achieved with no loss of performance. SSHN achieves similar performance to the best baseline with $\varphi = 0.15$, while reducing the computational cost of inference by 50%. Characterizing the accuracy-efficiency Pareto frontier using SSHN requires substantially fewer GPU-hours than the traditional approach, while enabling a substantially more fine-scale curve. In our experiments, the set of baselines required over an order of magnitude ($10\times$) longer to train compared to the single hypernetwork model, even when we employ a coarse-grained baseline search.

### 5.2 Analysis Experiments

In this section we perform additional experiments designed to shed light on why the hypernetwork models achieve higher test accuracy than the baseline models.

**Varying Prior Width**. We first study the quality of hypernetwork segmentation results with varying width of the prior distribution $p(\varphi)$. Our goal is to understand how learning with varying amounts of rescaling affects segmentation accuracy and generalization. We train a set of SSHN models on the OASIS dataset but with narrow uniform distributions $\mathcal{U}(0.5 - r, 0.5 + r)$, where $r$ controls the width of the distribution. We vary $r = [0, 0.01, \ldots, 0.05]$ and evaluate them with $\varphi = 0.5$ Figure 8 shows Dice scores for training and test set on the OASIS segmentation task, as well as a baseline UNet for comparison. With $r = 0$, the hypernetwork is effectively trained with a constant $\varphi = 0.5$, and slightly underperforms compared to the baseline UNet. However, as the range of the prior distribution grows, the test Dice score of the hypernetwork models improves, surpassing the baseline UNet performance.

Figure 8 suggests that the improvement over the baselines can be at least partially explained by the hypernetwork learning a more robust model because it is exposed to varying amounts of feature rescaling. This suggests that stochastically changing the amount of downsampling at each iteration has a regularization effect on learning that leads to improvements in the model generalization. Wider ranges of rescaling values for the prior $p(\varphi)$ have a larger regularization effect. At high values of $r$, improvements become marginal and the accuracy of a model trained with the widest interval $p(\varphi) = \mathcal{U}(0.4, 0.6)$ is similar to the results of the previous experiments with $p(\varphi) = \mathcal{U}(0, 1)$.

**Weight transferability.** We study the effect of using a set of weights that was trained with a different rescaling factor. Our goal is to understand how the hypernetwork adapts its predictions as the rescaling factor changes. We use *fixed* weights $\theta' = h(0.5; \omega)$ as predicted by the SSHN model with input $\varphi' = 0.5$, and run inference using different rescaling factors $\varphi$. For comparison, we apply the same procedure to a set of weights of a Fixed U-Net baseline trained with a fixed $\varphi = 0.5$. Figure 9 presents segmentation results on OASIS for varying rescaling factors $\varphi$. Transferring weights from the baseline model leads to a rapid decrease in performance away from $\varphi = 0.5$. In contrast, the weights generated by the hypernetwork are more robust and can effectively be used in the range $\varphi \in [0.45, 0.6]$. Nevertheless, weights learned for a specific rescaling factor $\varphi$ do not generalize well

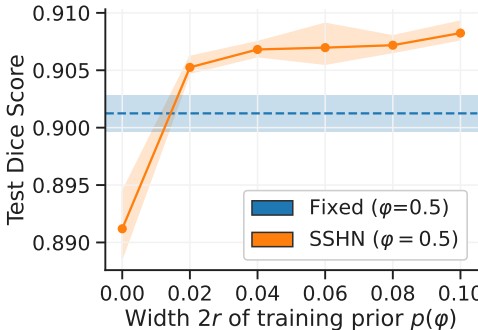

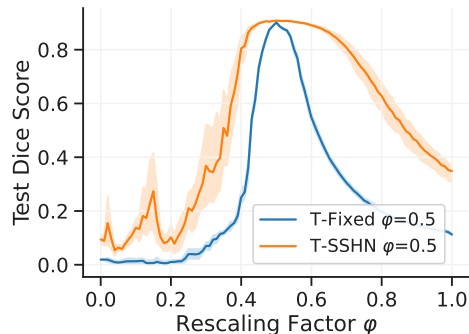

Figure 8: Segmentation Dice on OASIS Test data as a function of the prior distribution range $2r$. Results are evaluated with $\varphi = 0.5$ in all settings, regardless of the training prior, and we include a baseline UNet (Fixed) model. Uniform priors for $\varphi$ are of the form $\mathcal{U}(0.5 - r, 0.5 + r)$, i.e. they linearly increase around 0.5. Standard deviation (shaded regions) is computed over 5 random initializations.

Figure 9: Segmentation results for models resulting from transferring weights learned for a given rescaling factor and evaluated at different factors. The convolutional parameters learned by the hypernetwork (T-SSHN) transfer to nearby factors substantially better than the weights of the baseline model (T-Fixed). Standard deviation (shaded regions) is computed over 5 random seeds.

to a wide range of different rescaling factors, whereas weights predicted by the hypernetwork transfer substantially better to other rescaling factors.

## 6  Limitations

We evaluated our proposed framework on image segmentation and registration, a prevalent tasks in medical image analysis. We believe that our method can be extended to other tasks like object detection or classification. Constructing such extensions is an exciting area for future research. Furthermore, we focus on the U-Net as a benchmark architecture because it is a popular choice for segmentation networks across many domains [58, 15, 32, 31, 2, 4]. It also features key properties, such as an encoder-decoder structure and skip connections from the encoder to the decoder, that are used in other segmentation architectures [8, 74, 9]. Nevertheless, in Section C.1 of the Supplemental material, we report results for other popular segmentation architectures, including PSPNet, FPN and SENet, and find analogous results.

## 7  Conclusion

We introduce SSHN: a hypernetwork-based approach for efficiently learning a family of CNNs with different rescaling factors in an amortized way. Given a trained model, SSHN enables efficiently generating the Pareto frontier for the trade-off between inference cost and accuracy. This makes it possible to rapidly search for a rescaling factor that maximizes accuracy subject to computational cost constraints. For example, using the hypernetwork, we demonstrate that using larger amounts of downsampling can often lead to a substantial reduction in the computational costs of inference, while sacrificing nearly no accuracy. Finally, we demonstrate that for a variety of biomedical image analysis tasks and datasets, SSHN models models achieve accuracy at least as good, and more often better than, CNNs trained with fixed rescaling ratios, while producing more robust models. This strategy enables the construction of a new class of cost-adjustable models, which can be adapted to the resources of the user and minimize carbon footprint while providing competitive performance in biomedical image analysis.

## Acknowledgements

This research is supported by Quanta Computer, Inc, and Wistron Coporation. Additionally, the project was supported by NIH R01AG064027 and R01AG070988.

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
