# A   Experimental Setup

## A.1   Implementation Details

We provide additional implementation details:

**Architecture**. To bridge the mismatch between continuous rescaling factor $\varphi$ values and discrete spatial dimensions in the feature maps, we round the spatial dimensions of the outputs of downscaling operations to the nearest integer value. For upscaling operations we use $1/\varphi$ but round the upscaled spatial dimensions so that they match the dimensions of the features from the skip connections used in the concatenation. We use a final convolution, with kernel size 1x1 and without activation function, to transform the number of channels to the target number of labels for the segmentation task.

**Initialization**. We separate the weights of the last fully connected layer of the hypernetwork into different groups, each one corresponding to a fully connected layer predicting an individual parameter tensor of the primary network. This strategy leads to each predicted parameter of the primary network having an initialization that only depends on its own dimensions and not on the entire architecture.

**Training**. We use the Adam optimizer with default $\beta$ parameters $\beta_1 = 0.9$ and $\beta_2 = 0.99$. For training we include a label for the background to ensure the cross-entropy loss has exactly one label for each pixel in the output. We do not use the background label when computing evaluation metrics.

# B  Additional Results

Table 1: Segmentation performance on each dataset in terms of mean Dice score for a held-out test set, across five random seeds, as well as the corresponding computational cost of running inference (in GFLOPs). We report mean Dice score over the anatomical structures and standard deviation across model initialization.

| Dataset | $\varphi$ | GFLOPs | Stochastic | FiLM | Fixed | SSHN |
|---------|-----------|--------|------------|------|-------|------|
| CAMUS | 0.00 | 0.47 | $30.5 \pm 1.0$ | $32.2 \pm 1.1$ | $39.1 \pm 0.7$ | $35.8 \pm 0.6$ |
| | 0.05 | 0.47 | $66.1 \pm 0.5$ | $67.3 \pm 0.7$ | $69.1 \pm 0.6$ | $72.4 \pm 0.4$ |
| | 0.10 | 0.49 | $74.3 \pm 0.5$ | $73.0 \pm 1.8$ | $77.5 \pm 0.1$ | $80.0 \pm 0.3$ |
| | 0.15 | 0.50 | $77.3 \pm 0.4$ | $77.6 \pm 0.6$ | $79.4 \pm 0.5$ | $82.1 \pm 0.6$ |
| | 0.20 | 0.53 | $79.0 \pm 0.3$ | $79.2 \pm 1.1$ | $77.5 \pm 0.5$ | $83.9 \pm 0.3$ |
| | 0.25 | 0.57 | $80.7 \pm 0.4$ | $80.0 \pm 1.0$ | $80.7 \pm 0.2$ | $85.0 \pm 0.3$ |
| | 0.30 | 0.62 | $81.4 \pm 0.2$ | $80.3 \pm 2.7$ | $79.4 \pm 0.2$ | $84.9 \pm 0.3$ |
| | 0.35 | 0.69 | $82.4 \pm 0.3$ | $81.5 \pm 2.0$ | $79.6 \pm 0.6$ | $85.3 \pm 0.4$ |
| | 0.40 | 0.77 | $82.5 \pm 0.6$ | $82.4 \pm 1.0$ | $81.8 \pm 0.3$ | $86.0 \pm 0.2$ |
| | 0.45 | 0.89 | $83.1 \pm 0.5$ | $82.5 \pm 1.3$ | $82.5 \pm 0.3$ | $86.0 \pm 0.1$ |
| | 0.50 | 1.07 | $83.3 \pm 0.1$ | $83.2 \pm 0.3$ | $83.5 \pm 0.3$ | $86.2 \pm 0.3$ |
| OASIS | 0.00 | 0.47 | $12.1 \pm 0.7$ | $13.6 \pm 1.4$ | $31.5 \pm 0.5$ | $28.3 \pm 0.7$ |
| | 0.05 | 0.47 | $60.9 \pm 3.0$ | $61.0 \pm 5.8$ | $80.5 \pm 0.5$ | $82.7 \pm 0.1$ |
| | 0.10 | 0.49 | $52.3 \pm 1.8$ | $61.8 \pm 6.0$ | $84.9 \pm 0.5$ | $86.7 \pm 0.3$ |
| | 0.15 | 0.50 | $81.6 \pm 0.3$ | $83.2 \pm 1.2$ | $86.9 \pm 0.2$ | $89.0 \pm 0.2$ |
| | 0.20 | 0.53 | $81.4 \pm 1.6$ | $84.1 \pm 1.5$ | $87.9 \pm 0.4$ | $89.6 \pm 0.1$ |
| | 0.25 | 0.57 | $83.9 \pm 0.1$ | $85.8 \pm 0.5$ | $88.2 \pm 0.1$ | $90.2 \pm 0.2$ |
| | 0.30 | 0.62 | $84.8 \pm 0.3$ | $86.5 \pm 0.7$ | $88.3 \pm 0.3$ | $90.3 \pm 0.1$ |
| | 0.35 | 0.69 | $83.9 \pm 0.7$ | $86.4 \pm 0.8$ | $88.6 \pm 0.2$ | $90.5 \pm 0.1$ |
| | 0.40 | 0.77 | $84.7 \pm 0.5$ | $87.0 \pm 0.5$ | $88.8 \pm 0.2$ | $90.6 \pm 0.1$ |
| | 0.45 | 0.89 | $84.2 \pm 0.5$ | $86.5 \pm 0.4$ | $88.9 \pm 0.1$ | $90.6 \pm 0.1$ |
| | 0.50 | 1.07 | $84.8 \pm 0.2$ | $87.2 \pm 0.6$ | $89.1 \pm 0.0$ | $90.7 \pm 0.1$ |
| PanDental | 0.00 | 0.47 | $56.0 \pm 1.7$ | $60.2 \pm 0.9$ | $68.5 \pm 2.0$ | $63.7 \pm 0.8$ |
| | 0.05 | 0.47 | $80.7 \pm 0.6$ | $80.5 \pm 0.4$ | $83.8 \pm 0.3$ | $83.9 \pm 0.4$ |
| | 0.10 | 0.49 | $80.1 \pm 0.9$ | $79.7 \pm 0.9$ | $85.2 \pm 0.2$ | $85.0 \pm 0.7$ |
| | 0.15 | 0.50 | $81.9 \pm 0.5$ | $82.1 \pm 0.8$ | $86.5 \pm 0.4$ | $86.4 \pm 0.6$ |
| | 0.20 | 0.53 | $84.1 \pm 0.7$ | $84.6 \pm 0.4$ | $85.8 \pm 0.8$ | $87.5 \pm 0.8$ |
| | 0.25 | 0.57 | $84.7 \pm 0.8$ | $85.2 \pm 0.3$ | $87.6 \pm 0.1$ | $88.6 \pm 0.5$ |
| | 0.30 | 0.62 | $85.1 \pm 0.5$ | $85.5 \pm 0.5$ | $87.3 \pm 0.0$ | $88.9 \pm 0.3$ |
| | 0.35 | 0.69 | $85.6 \pm 0.9$ | $85.7 \pm 0.4$ | $86.9 \pm 0.1$ | $89.4 \pm 0.2$ |
| | 0.40 | 0.77 | $85.7 \pm 0.9$ | $85.5 \pm 0.4$ | $87.9 \pm 0.1$ | $89.7 \pm 0.2$ |
| | 0.45 | 0.89 | $85.8 \pm 0.7$ | $85.7 \pm 0.6$ | $88.1 \pm 0.1$ | $89.8 \pm 0.2$ |
| | 0.50 | 1.07 | $85.6 \pm 0.7$ | $85.4 \pm 0.6$ | $88.9 \pm 0.2$ | $89.9 \pm 0.3$ |
| WBC | 0.00 | 0.47 | $68.9 \pm 1.3$ | $73.1 \pm 1.2$ | $76.6 \pm 0.9$ | $75.9 \pm 1.8$ |
| | 0.05 | 0.47 | $85.6 \pm 0.1$ | $86.1 \pm 0.5$ | $86.5 \pm 0.5$ | $85.8 \pm 2.0$ |
| | 0.10 | 0.49 | $84.3 \pm 0.2$ | $84.9 \pm 1.3$ | $85.9 \pm 0.8$ | $88.4 \pm 0.6$ |
| | 0.15 | 0.50 | $85.5 \pm 0.9$ | $83.7 \pm 2.1$ | $86.9 \pm 0.4$ | $89.1 \pm 0.3$ |
| | 0.20 | 0.53 | $86.7 \pm 1.2$ | $87.0 \pm 1.2$ | $86.6 \pm 0.5$ | $90.6 \pm 0.1$ |
| | 0.25 | 0.57 | $88.7 \pm 0.4$ | $87.5 \pm 1.0$ | $88.3 \pm 0.6$ | $91.8 \pm 0.1$ |
| | 0.30 | 0.62 | $88.7 \pm 0.5$ | $87.2 \pm 0.7$ | $87.5 \pm 0.7$ | $91.8 \pm 0.2$ |
| | 0.35 | 0.69 | $89.0 \pm 0.7$ | $87.6 \pm 0.8$ | $86.7 \pm 0.4$ | $92.2 \pm 0.1$ |
| | 0.40 | 0.77 | $87.7 \pm 1.8$ | $87.5 \pm 0.8$ | $89.1 \pm 0.3$ | $92.7 \pm 0.1$ |
| | 0.45 | 0.89 | $88.3 \pm 1.6$ | $86.5 \pm 0.9$ | $89.3 \pm 0.2$ | $92.7 \pm 0.4$ |
| | 0.50 | 1.07 | $88.1 \pm 2.0$ | $87.1 \pm 1.4$ | $90.9 \pm 0.6$ | $92.7 \pm 0.4$ |

Table 2: Registration performance on the OASIS dataset in terms of mean Dice score for a held-out test set, across five random seeds, as well as the corresponding computational cost of running inference (in GFLOPs). We report mean Dice score over the anatomical structures and standard deviation across model initialization.

| $\varphi$ | GFLOPs | Stochastic | FiLM | Fixed | SSHN |
|---|---|---|---|---|---|
| 0.00 | 0.47 | $66.6 \pm 0.1$ | $67.4 \pm 0.4$ | $67.7 \pm 0.2$ | $67.7 \pm 0.2$ |
| 0.05 | 0.47 | $66.9 \pm 0.2$ | $67.6 \pm 0.3$ | $67.9 \pm 0.2$ | $67.8 \pm 0.2$ |
| 0.10 | 0.49 | $67.2 \pm 0.1$ | $67.8 \pm 0.4$ | $68.5 \pm 0.1$ | $68.4 \pm 0.3$ |
| 0.15 | 0.50 | $67.6 \pm 0.3$ | $68.6 \pm 0.5$ | $69.1 \pm 0.2$ | $69.1 \pm 0.2$ |
| 0.20 | 0.53 | $68.6 \pm 0.0$ | $69.5 \pm 0.4$ | $69.5 \pm 0.0$ | $69.9 \pm 0.2$ |
| 0.25 | 0.57 | $69.1 \pm 0.0$ | $70.0 \pm 0.4$ | $70.0 \pm 0.1$ | $70.5 \pm 0.3$ |
| 0.30 | 0.62 | $69.5 \pm 0.1$ | $70.3 \pm 0.4$ | $70.2 \pm 0.1$ | $70.9 \pm 0.3$ |
| 0.40 | 0.77 | $70.0 \pm 0.2$ | $70.5 \pm 0.4$ | $70.7 \pm 0.1$ | $71.5 \pm 0.3$ |
| 0.45 | 0.89 | $70.2 \pm 0.2$ | $70.6 \pm 0.5$ | $71.2 \pm 0.2$ | $71.6 \pm 0.3$ |
| 0.50 | 1.07 | $70.3 \pm 0.2$ | $70.7 \pm 0.5$ | $71.5 \pm 0.1$ | $71.7 \pm 0.3$ |

Table 3: Training time (in minutes) for the each method and task. For the Fixed baseline we take into account the time taken to train all the independent models with different scale factors necessary to characterize the Pareto frontier. We find that Stochastic, FiLM and SSHN all have similar training times whereas the set of Fixed models requires substantially more time to train.

| Task | Dataset | Stochastic | FiLM | Fixed | SSHN |
|---|---|---|---|---|---|
| Registration | OASIS | 174.3 | 182.6 | 1867.7 | 208.8 |
| Segmentation | CAMUS | 48.8 | 50.5 | 512.1 | 57.0 |
| | OASIS | 255.1 | 264.3 | 2811.1 | 304.5 |
| | PanDental | 48.7 | 46.9 | 485.0 | 54.3 |
| | WBC | 40.3 | 45.5 | 449.8 | 47.3 |

# C   Additional Experimental Results

## C.1   Segmentation Architecture Ablation

We originally tested our method on using the U-Net architecture [58], which is a popular architecture choice in image segmentation tasks, particularly for medical imaging applications [29]. Nevertheless, our method can be extended to other architectures. Given an arbitrary segmentation network, our method requires replacing fixed resizing layers with variable resizing layers and using the hypernetwork to predict all the convolutional parameters of the network.

In this experiment we test our method on other popular alternative segmentation architectures. While the majority of segmentation architectures feature the same basic components (convolutional layers, resizing layers, skip connections), we aimed to cover other architectual components not included in the U-Net design such as pyramid spatial pooling layers and squeeze-excitation layers. The ablation includes the following architectures:

- **Residual UNet**. This is popular U-Net variant where the operations at each resolution feature a residual connection similar to the ResNet architecture [62]. Previous work has highlighted the importance of this type of connections in biomedical image segmentation [14].

- **FPN**. Feature Pyramid Networks [44] construct a pyramid of features at several resolution levels, combining them in a fully convolutional manner and performing predictions at multiple resolutions during training.

- **PSPNet**. Pyramid Spatial Pooling [75] layers gather context information at multiple resolution levels in parallel, and several competitive natural image segmentation models include them in their architectures [9]. For our implementation, we perform the dynamic resizing operations in the encoder part and used the multi resolution PSP blocks in the decoder part.

- **SENet**. Squeeze Excitation networks [26] incorporate an attention mechanism to dynamically reweigh feature map channels during the forward pass. Squeeze-Excitation layers have been shown to be performant in segmentation tasks [52, 59, 77]. In our implementation, we include Squeeze-Excitation layers to both the encoder and the decoder stages of the network.

For each considered architecture we compare a single hypernetwork with a variable rescaling factor to a set of individual baselines with varying rescaling factors. We train baselines at 0.05 $\varphi$ increments, i.e. $\varphi = 0, 0.05, 0.1, \ldots, 0.5$. We evaluate on the OASIS2d semantic segmentation task introduced in the paper, which features 24 brain structure labels. We evaluate using Dice score and average over the labels. Results in Figure 10 show segmentation quality as the rescaling factor $\varphi$ varies. We find similar results to those in the paper (Figure 3) with our single hypernetwork model matching the set of individually trained networks.

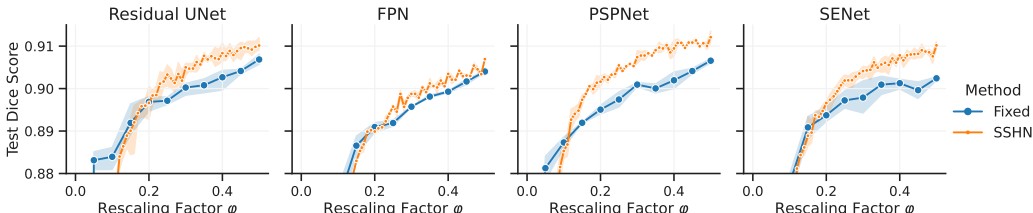

Figure 10: **Segmentation Architecture Ablation** Test Dice Score on the OASIS segmentation task for various choices of primary network architecture. Each plot features a family of networks trained with fixed amounts of feature rescaling (Fixed) and a single instance of the proposed hypernetwork method (SSHN) trained on the entire range of rescaling factors. Results are averaged across three random initializations, and the shaded regions indicate the standard deviation across them.

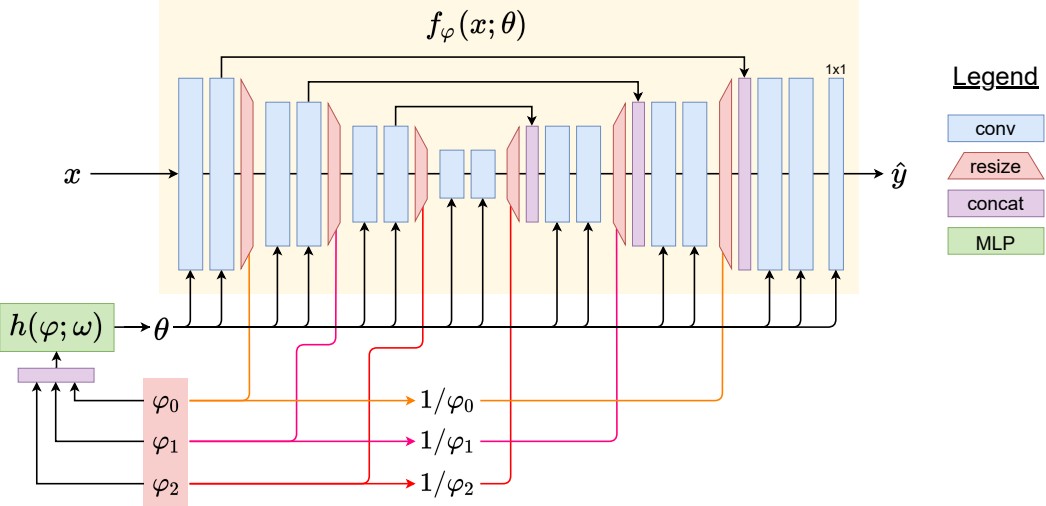

Figure 11: Diagram for Scale-Space Hypernetworks with multiple separate rescaling factors $(\varphi_0, \varphi_1, \varphi_2)$. Each rescaling factor $\varphi_i$ determines the downscaling and upscaling of the $i^{\text{th}}$ downscaling step and the $i^{\text{th}}$ to last upscaling step. Given a series of rescaling factor values $\boldsymbol{\varphi}$, the hypernetwork $h(\varphi; \omega)$ predicts the parameter values $\theta$ of the primary segmentation network $f_\varphi(x; \theta)$.

## C.2    Separate Rescaling Factors

In the main body of the paper, we explored using a single rescaling factor $\varphi$ to control all rescaling operations in the network. In this section, we also study a more general formulation where each downscaling step is controlled by a separate rescaling factor $\varphi_i$. Our experiments show that a hypernetwork model is capable of modeling this hyperparameter space despite the increased dimensionality. We find that this more general formulation achieves comparable accuracy and efficiency to the single rescaling factor model, with similar accuracy-efficiency Pareto frontiers.

**Method**. We use separate factors $\boldsymbol{\varphi} = \{\varphi_0, \varphi_1, \ldots, \varphi_K\}$ for each rescaling operation in the network. We train the hypernetwork $h(\varphi; \omega)$ to predict the set of weights $\theta$ for a set of rescaling factors $\boldsymbol{\varphi}$. For a network with $K$ downscaling and $K$ upscaling steps, the rescaling factor $\varphi_i$ controls both the downscaling of the $i^{\text{th}}$ block of the network and the upscaling of the $(K-i)^{\text{th}}$ upscaling block to achieve consistent spatial dimensions between the respective encoder and decoder. Figure 11 presents a diagram of the flow of information for the case of a UNet with four levels and three rescaling steps.

**Setup**. We train the hypernetwork model on the OASIS segmentation task using the same experimental setup as the single rescaling factor experiments. We sample each $\varphi_i \sim \mathcal{U}(0, 1)$ independently. We use the same $(\varphi_i, 1 - \varphi_i)$ input encoding to prevent biasing the predicted weights towards the magnitude of the input values. Once trained, we evaluate using the test set, sampling each of the three rescaling factors by 0.1 increments. This amounts to $11^3 = 1331$ different settings.

**Results**. Figure 12 shows Dice scores for the OASIS test set as we vary each of the rescaling factors $\varphi$ in the network. The hypernetwork is capable of achieving high segmentation accuracy ($> 0.88$ dice points) for the vast majority of $\boldsymbol{\varphi} = (\varphi_0, \varphi_1, \varphi_2)$ settings. Figure 13 shows the trade-off between segmentation accuracy and inference cost for the evaluated settings of $\boldsymbol{\varphi}$ along with the baseline and the hypernetwork model with a single rescaling factor.

**Discussion**. We find that first rescaling factor $\varphi_0$ has the most influence of segmentation results, while $\varphi_2$ has the least. The model achieves optimal results when $\varphi_0 \in [0.3, 0.5]$. We observe that as either $\varphi_0$ or $\varphi_1$ approach zero, segmentation accuracy sharply drops, consistent with the behavior we found in the networks with a common rescaling factor. Having separate rescaling factors $\boldsymbol{\varphi}$ yields results comparable to using a single factor $\varphi$ for the whole network.

We highlight that $h(\boldsymbol{\varphi}; \omega)$ manages to learn a higher dimensional hyperparameter input space successfully using the same model capacity. Moreover, we find that for the vast majority of $\boldsymbol{\varphi}$ settings the segmentation accuracy is quite high for the segmentation task, suggesting that the task can be

solved with a wide array of rescaling settings. In Figure 13 we show that despite the increased hyperparameter space the Pareto frontier closely matches the results of using a single rescaling factor, suggesting that having a single rescaling factor is a good choice for this task.

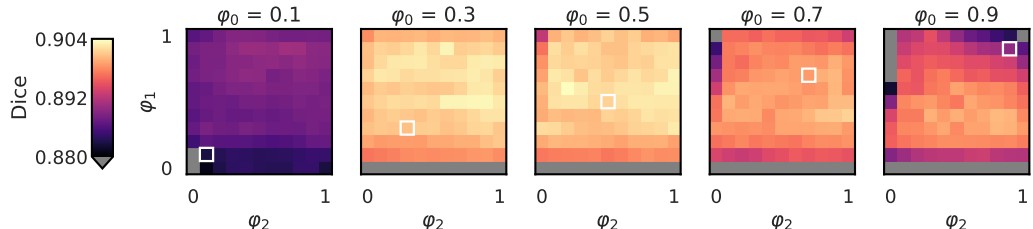

Figure 12: Segmentation results on a hypernetwork model with separate factors $\varphi = (\varphi_0, \varphi_1, \varphi_2)$ for each rescaling operation in the segmentation network. We report mean dice score across all anatomical structures on the OASIS test set. We highlight in white the $\varphi$ settings explored by our single factor model. We mask underperforming settings (<0.88) with grey to maximize the dynamic range of the color map. We find that the first rescaling factor $\varphi_0$ has a substantially larger influence in segmentation quality compared to $\varphi_1$ and $\varphi_2$.

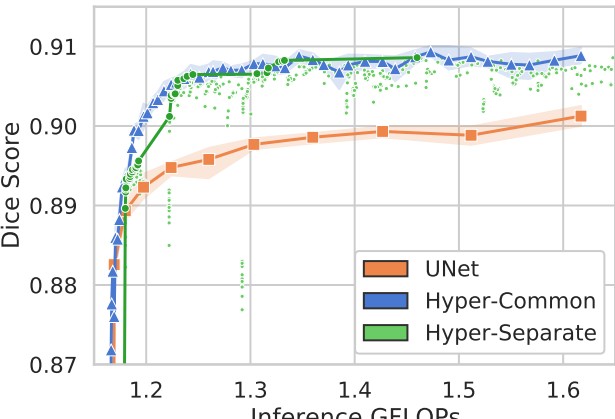

Figure 13: Segmentation accuracy as a function of required GFLOPs for one inference pass in the network. Mean Dice score across all anatomical structures on the OASIS test set for the baseline models (UNet), the hypernetwork model with a common rescaling factor $\varphi$ (Hyper-Common), and the hypernetwork model with separate rescaling factors $(\varphi_0, \varphi_1, \varphi_2)$. For Hyper-Separate we include both individual suboptimal points and the Pareto frontier as a line.

# D  Additional Analysis

We include additional visualizations of how varying the rescaling factor $\varphi$ affects the predicted weights and the associated feature maps. Figure 15 illustrates the predicted network parameters for a series of channels and layers as the rescaling factor $\varphi$ varies. To capture the variability of the predicted parameters, Figure 14 presents the coefficient of variation for parameters across a series of rescaling factors. We observe that the first layer presents little to no variation as $\varphi$ changes. Figure 16 presents sample feature maps as the rescaling factor $\varphi$ changes, for separate layers in the network. The predicted convolutional kernels by the hypernetwork extract similar anatomical regions for different values of $\varphi$.

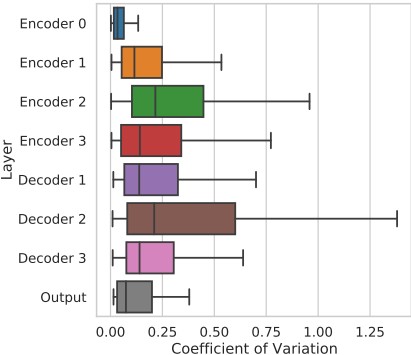

Figure 14: Coefficient of variation (CV) ($\sigma/|\mu|$) for each convolutional parameter predicted by the hypernetwork across a series of rescaling ratios $\varphi = \{0, 0.1, 0.2, \ldots, 1.0\}$. Features at the first layer and the last layer present substantially smaller CV values compared to the rest of the layers.

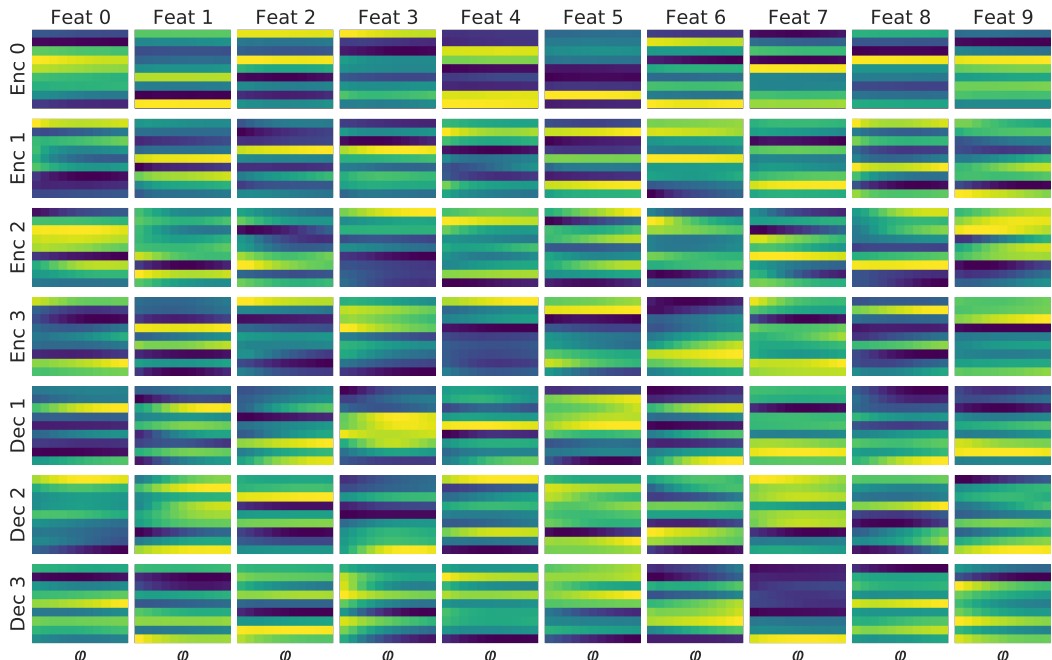

Figure 15: Sample predicted convolutional parameters maps for various features channels (columns) and for different primary network layers (rows). For each feature we show how each convolutional parameter (y-axis) changes as the rescaling factor $\varphi$ varies (x-axis). We find that the hypernetwork leads to substantial variability as $\varphi$ changes in some layers, especially those near the middle of the architecture.

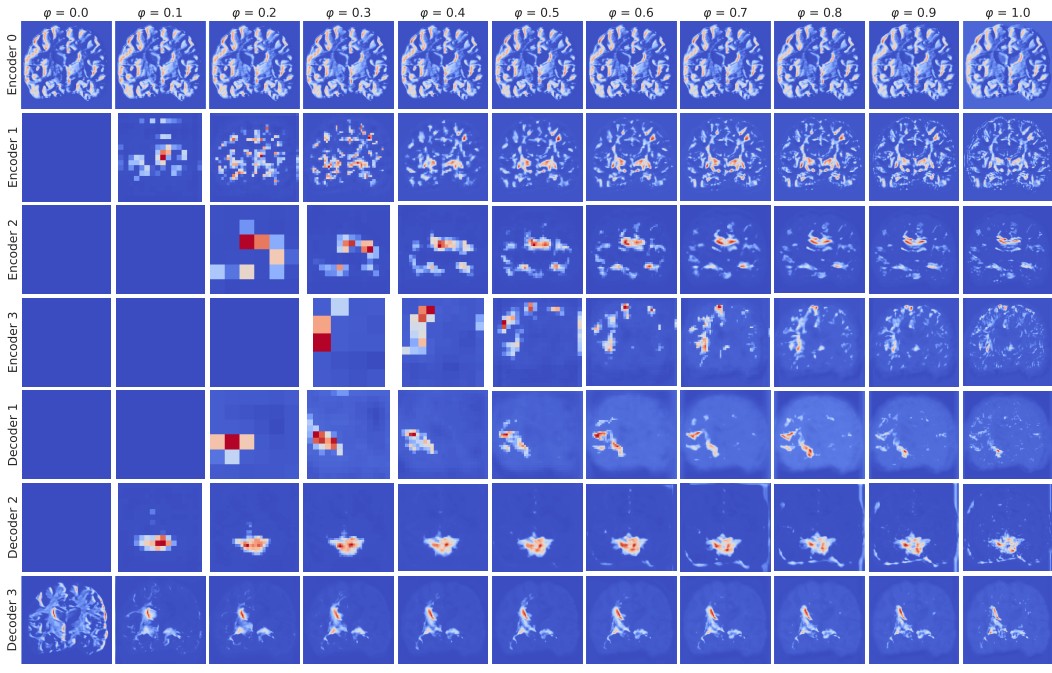

Figure 16: Sample intermediate convolutional feature maps for various rescaling factors $\varphi$ (columns) and different primary network layers (rows).