# OpenReview forum: "Scale-Space Hypernetworks for Efficient Biomedical Image Analysis"
_NeurIPS.cc/2023/Conference — NeurIPS 2023 poster_

### Official Review · Reviewer_VkYJ · 2023-06-13

**Soundness:** 2 fair
**Presentation:** 2 fair
**Contribution:** 2 fair
**Rating:** 3
**Confidence:** 5

**Summary:**

The authors propose a unified approach based on Hypernetworks (HN) to model the accuracy-efficiency pareto front for medical applications.
The authors claim the following contributions:
- Introducing Scale-Space HyperNetworks (SSHN) a single model that given a rescaling factor generates weights for corresponding model with reduced spatial dimensions of intermediate features.
- Demonstrating the performance of SSHN on varing datasets while reducing the number of FLOPs by up to 50%.


**Strengths:**

- The paper is well-written and easy to follow.
- It tackles an important problem of computational heterogeneity during inference.
- The proposed method is simple and easy to implement/reproduce.


**Weaknesses:**

- The authors failed to cite pioneering HN works [1,2,3]. Specifically [1] which is highly relevant and related to this work.
- The novelty of this paper is limited. Pareto Front Learning (PFL) [1] is a widely explored field within machine learning, encompassing various lines of research. It has gained significant recognition and has a well-established presence due to its diverse range of investigations and studies. PFL is a computational approach that aims to find the optimal trade-offs between multiple conflicting objectives. It involves identifying a set of solutions that lie on the Pareto front, representing the best possible outcomes for each objective, without sacrificing performance in other areas. By exploring the Pareto front, decision-makers can make informed choices that balance competing objectives and achieve more comprehensive and balanced solutions. It seems like the paper’s core novelty has already been presented by previous works. I suggest the authors to better explain how their approach differs from highly related works in PFL [1,4].
- I wonder how well the proposed approach works on non-medical datasets e.g. Cityscapes, NYU, etc.
- How well the model generalizes to unseen scales i.e. scales that are not in the range of $p(\varphi)$?
- I suggest the authors add a section regarding their design choices for example (i) the prior scaling factor and (ii) the rescale module $R_\varphi$. Ablation experiments should be included as well in the main manuscript or supplementary.

Citations:

[1] Learning the Pareto Front with Hypernetworks, Navon et al.

[2] HyperStyle: StyleGAN Inversion with HyperNetworks for Real Image Editing, Alaluf et al.

[3] Personalized Federated Learning using Hypernetworks, Shamsian et al.

[4] Controllable Pareto Multi-Task Learning, Lin et al.


**Questions:**

- Line 105 - instead of $(C,\lceil\varphi H\rceil,\lceil \varphi H \rceil)$ should be $(C,\lceil\varphi H\rceil,\lceil \varphi W \rceil)$.
- Line 128-129 adding $R_\varphi$ will make it clearer for the reader, in addition, consider moving these lines to ~Line 105 where you first present the notation $R_\varphi$.
- Why not normalizing $\varphi$ such that $[0, 0.5] \rightarrow [0,1]$ this will increase $\varphi$’s resolution.
- Did you try to use different prior over $\varphi$, $p(\varphi)$ instead of uniform?


**Limitations:**

See weaknesses and questions.

---

> ### Author Rebuttal · Authors · 2023-08-09
>
>
> We appreciate the constructive feedback and comments. Addressing the
> raised questions:
>
> ### Weaknesses:
>
> -   **Prior Work** - We agree that the work cited in the review is
>     relevant to our method, and we will revise our manuscript to draw
>     the connections and contextualize the research.
>
> -   **Pareto Front Learning** - The goal of our work is substantially
>     different from that of the referenced work \[1\]. In particular, PFL
>     focuses on jointly optimizing the Pareto frontier of multiple
>     predetermined training objectives across a single architectural
>     setting. In contrast, our approach is about optimizing a single
>     objective (in our case segmentation accuracy) across a range of
>     architectural settings that introduce various computational
>     trade-offs. To the best of our knowledge, our work is the first to
>     explore amortizing the cost of training models with different
>     internal rescaling factors, a key aspect in convolutional model
>     efficiency, and that has not been explored in any of the references.
>     The technical obstacles associated with our goal are different from
>     those faced by PFL. For us, metrics of interest like computational
>     cost (FLOPs), required memory, or model latency are not
>     differentiable, and thus cannot be easily incorporated in the
>     learning process in the way the cited work does. Our approach allows
>     practitioners to use the learned hypernetwork to efficiently
>     generate the accuracy-cost frontier for their given hardware
>     constraints, and then choose the model to be deployed based on
>     validation metrics that do not need to be defined during the
>     training process.
>
> -   **Non-medical Datasets** - Our work focuses on medical datasets
>     since in this domain it is common to train models from scratch,
>     which is often computationally intensive. In contrast, natural image
>     segmentation models typically use a pretrained encoder backbone trained at a fixed
>     set of resolutions.
>
> -   **Scaling shift** - Our model performs best when evaluated with
>     rescaling factors within $p(\varphi)$. Outside this range
>     segmentation performance degrades. However, the assumption is that
>     if a scale is of interest it should be in the range of $p(\varphi)$.
>
> -   **Ablations** - We include ablation experiments regarding the model
>     architecture in Section C.1 of the manuscript. We will provide
>     further design details regarding the resizing module $R_\varphi$.
>     However, we are unsure about what alternative choices of rescale
>     module  $R_\varphi$ would be of interest. We believe that bilinear
>     interpolation is the most prevalent fractional resizing mechanism in
>     the literature.
>
> ### Questions:
>
> -   We are thankful for the corrections, and we will revise the
>     manuscript to include them.
>
> -   We do normalize $[0, 0.5] \rightarrow [0, 1]$ and then apply a
>     $(x, 1-x)$ encoding. We will revise our implementation details.
>
> -   We did explore using an *area uniform* prior
>     $p(\varphi) \sim \sqrt{\mathcal{U}(0,1)}$ but we did not find any
>     significant differences in the produced Pareto curves.

---

> > ### Comment · Reviewer_VkYJ · 2023-08-12
> > **Reviewer response**
> >
> > - **Non-medical Datasets** - I tend to disagree with the authors. In bio-medical as in other domains is all about personalization. This need comes up in different domains hence involving various datasets. When proposing a new approach, we as a community want to see that is scales through different domains, datasets, and learning setups.
> >
> > - **Scaling shift** - Can the authors share results on $p(\varphi)$ that is out of range? i.e. extrapolation.
> >
> > - Another question that came to my mind is in terms of the number of learned parameters and latency. Can the authors elaborate on the overhead of HN in terms of latency (FLOPs, Wall-time) and learned parameters? also include these stats for the baselines please.

---

> > > ### Author Response · Authors · 2023-08-18
> > >
> > > -   **Non-medical datasets** - We agree that something that is universal
> > >     is better than something that is not. However, we highlight that
> > >     *biomedical image analysis* is its own substantial research field
> > >     that includes a broad variety of domains. For example, dental x-rays
> > >     are quite different from brain MRIs. Our experiments include results
> > >     spanning several imaging modalities and different anatomical
> > >     structures.
> > >
> > > -   **Scaling Shift** - The table below presents out-of-distribution
> > >     values of the rescaling factor ($\varphi > 0.5$) for our method
> > >     trained with $p(\varphi) = \mathcal{U}(0,0.5)$ on all the
> > >     segmentation problems. Dice score deteriorates as we increase the
> > >     rescaling factor. This makes sense because these factors were not
> > >     considered during training, and with lesser downsampling the model
> > >     cannot effectively aggregate visual context. In practice, all
> > >     scaling factors of interest should be included as part of
> > >     $p(\varphi)$ during training.
> > >
> > >
> > > |   $\varphi$ | CAMUS       | OASIS       | PanDental   | WBC         |
> > > |---------------:|:------------|:------------|:------------|:------------|
> > > |           0.50 | 0.90 (0.00) | 0.89 (0.00) | 0.94 (0.00) | 0.95 (0.00) |
> > > |           0.55 | 0.89 (0.00) | 0.89 (0.00) | 0.94 (0.00) | 0.94 (0.00) |
> > > |           0.60 | 0.88 (0.00) | 0.87 (0.00) | 0.93 (0.00) | 0.92 (0.00) |
> > > |           0.65 | 0.84 (0.00) | 0.83 (0.00) | 0.90 (0.00) | 0.90 (0.00) |
> > > |           0.70 | 0.77 (0.00) | 0.66 (0.03) | 0.86 (0.01) | 0.87 (0.01) |
> > > |           0.75 | 0.68 (0.01) | 0.44 (0.06) | 0.81 (0.01) | 0.84 (0.02) |
> > > |           0.80 | 0.61 (0.02) | 0.31 (0.05) | 0.78 (0.01) | 0.82 (0.03) |
> > > |           0.85 | 0.56 (0.02) | 0.25 (0.02) | 0.76 (0.01) | 0.79 (0.05) |
> > > |           0.90 | 0.51 (0.02) | 0.20 (0.01) | 0.75 (0.01) | 0.77 (0.06) |
> > > |           0.95 | 0.47 (0.02) | 0.16 (0.02) | 0.73 (0.01) | 0.74 (0.08) |
> > > |           1.00 | 0.42 (0.05) | 0.14 (0.02) | 0.70 (0.02) | 0.71 (0.09) |
> > >
> > >
> > > -   **Efficiency Measurements** - Tables 2 & 3 in the supplement present
> > >     inference and training costs, respectively. In Table 2 we present the
> > >     inference cost per rescaling factor and the corresponding accuracy
> > >     for our method and all baselines. In Table 3, we report the training
> > >     cost of characterizing the Pareto accuracy-efficiency frontier for
> > >     each method. In the table below, we report total parameter counts
> > >     (including hypernetwork weights and FiLM parameters) as well as the
> > >     number of primary network parameters for each method. We will revise
> > >     the manuscript to include the parameter counts.
> > >
> > >     In our experiments, training a SSHN model takes 10x less time than
> > >     training a set of Fixed baselines, and only 1.8x more costly than
> > >     training a single U-Net model. At inference time, the hypernetwork
> > >     is only used once to generate the weights of the primary network,
> > >     which is then used to make the predictions for different inputs, so
> > >     the computational cost is identical between SSHN and Fixed for a
> > >     given rescaling factor.
> > >
> > > | Method      | Total Params   | Primary Params   |
> > > |:------------|:---------------|:-----------------|
> > > | Fixed     | 109.4K         | 109.4K           |
> > > | Stochastic  | 109.4K         | 109.4K           |
> > > | FiLM        | 192.5K         | 109.4K           |
> > > | SSHN (ours) | 10.7M          | 109.4K           |

---

> > > > ### Comment · Reviewer_VkYJ · 2023-08-19
> > > > **Respond to authors**
> > > >
> > > > Upon realizing the disparity in parameter quantities, I am left to ponder if it could be the root of the issue. Did the authors perform a comparison where the number of parameters is identical between the methods?

---

> > > > > ### Author Response · Authors · 2023-08-19
> > > > >
> > > > > While a hypernetwork has more total parameters, those parameters are used to learn the landscape of primary network weights. The primary network, which solves the desired task, is only predicted once (for a given desired efficiency). At inference time, all methods have the same number of parameters.
> > > > >
> > > > > We chose the capacity of the primary network (109.4K parameters) architecture first, by performing a grid search on the Fixed baseline model size. Increasing the model size did not lead to improvements for the considered baselines.
> > > > >
> > > > > In our experiments, we keep the size of the primary network constant to ensure a fair comparison.

---

> > > > > > ### Comment · Reviewer_VkYJ · 2023-08-19
> > > > > > **Reviewer Response**
> > > > > >
> > > > > > I would like to thank the authors for their time and effort, I will keep my score unchanged.

---

### Official Review · Reviewer_Lhzq · 2023-07-05

**Soundness:** 3 good
**Presentation:** 4 excellent
**Contribution:** 3 good
**Rating:** 7
**Confidence:** 3

**Summary:**

This paper proposes to learn a spectrum for CNNs with varying internal rescaling factors and demonstrates the effectiveness of the proposed approach in several medical image analysis applications including segmentation and registration with fixed and dynamic rescaling factors. Overall, the approach is simple but sound and powerful, and the empirical results clearly support the claim. I recommend the paper for acceptance.

**Strengths:**

- Very clear presentation.
- Strong empirical results including one that nicely uncovers that many rescaling factors lead to similar results despite having substantially different inference costs for a variety of medical imaging tasks.
- Simple but powerful method that can characterize the trade-off between model accuracy and inference efficiency faster and better than the existing approaches.

**Weaknesses:**

n/a

**Questions:**

- How would your results change if you used an architecture more recent than U-Net for the tasks you evaluated on?
- Your proposed method seems to have a more smooth behavior compared to the other benchmarked resizing methods (Stochastic, FiLM) in Figure 3 -- do you have an intuition on why that might be?

**Limitations:**

Limitations are adequately addressed.

---

> ### Author Rebuttal · Authors · 2023-08-09
>
>
> We appreciate the constructive feedback and comments. Addressing the
> raised questions:
>
> -   **Primary Network Architecture** - We perform an architecture
>     ablation experiment in Section C.1 of the supplement. For the OASIS
>     dataset, we find similar trends to the ones of the U-Net
>     architecture.
>
> -   **Smooth behaviour** - We believe this is because the hypernetwork
>     formulation lets our model adapt better to changes in resolution. We
>     explored a related aspect in our *Weight transferability* experiment
>     in Section 5.2, where we find that SSHN weights transfer well to
>     neighboring rescaling factors.

---

> > ### Comment · Reviewer_Lhzq · 2023-08-18
> >
> > I have read the rebuttal by Authors and my assessment remains unchanged. Thanks for answering my questions.

---

### Official Review · Reviewer_wkRF · 2023-07-06

**Soundness:** 4 excellent
**Presentation:** 4 excellent
**Contribution:** 3 good
**Rating:** 7
**Confidence:** 4

**Summary:**

CNNs, particularly those handling 3D data, can pose computational challenges due to their high expense. To tackle this, researchers frequently scale down the input data, a practice that often compromises accuracy. This paper presents SSHN, a technique designed to learn a variety of CNN models, each with unique scaling factors. With a marginal increase in training duration, SSHN is demonstrated to enhance the trade-off between accuracy and efficiency, making it a promising solution to the limitations of conventional methods.

**Strengths:**

* The paper is well-written, with clear visuals and figure captions. The clarity of the text aids the understanding of the proposed concepts.
* The authors have chosen a reasonable variety of datasets for evaluation, strengthening the validity of their claims.
* Baseline approaches considered in the paper are relevant, providing a solid ground for comparison.
* The authors have diligently reported training five randomly initialized models, ensuring their findings are not reliant on a specific random seed.

**Weaknesses:**

* The paper states that CNNs are computationally intensive at inference time. However, in this reviewer's experience, the training time is often the more challenging aspect, especially with volumetric data. This warrants further clarification.
* The memory footprint of SSHN in the volumetric data regime is usually a big concern, as it often leads to memory consumption issues. It is understood that the authors did not investigate this aspect.
* It is unclear how the Fixed baseline was utilized during inference. For instance, it is not specified how the best rescaling factor is chosen for a specific dataset, or if the models are used as an ensemble.
* The handling of 3D datasets (such as OASIS and CAMUS) in experiments is unclear. It is uncertain whether they are treated as 2D datasets, and if the authors have considered at all using a 3D U-Net as their primary model. If datasets were used as 2D images, this must be mentioned in the limitations section.

**Questions:**

1. At the moment, training time has been only discussed in the supplementary material. Can you comment on the computational burden of SSHN and its implications? Would this warrant updating the limitations section?
2. Could you provide information on the memory footprint of training the SSHN in the volumetric data regime? Would it be feasible?
3. Could you elaborate on how the Fixed baseline is used during inference time and how the rescaling factor is chosen?
4. Have 3D datasets in your experiments been treated as 2D datasets? Have you considered using a 3D U-Net as the primary model?

**Limitations:**

The authors have discussed certain limitations including the choice of model architecture and tasks. However, the authors could further elaborate on how method's training time and memory footprint for volumetric data, as this could be a significant limitation for practical applications.

---

> ### Author Rebuttal · Authors · 2023-08-09
>
>
> We appreciate the constructive feedback and comments. Addressing the
> raised questions:
>
> 1.  **Training Cost** - We would like to clarify how the model
>     development process is performed with our method:
>
>     -   Training - The hypernetwork is trained by generating the primary
>         network weights from randomly sampled rescaling factors.
>
>     -   Scale selection - Once trained, the hypernetwork is used once per rescaling factor to predict the weights of each primary network. These weights are used to evaluate the accuracy on a held-out set of data, producing a Pareto accuracy-efficiency frontier.
>     A rescaling factor is then chosen based on the trade-off characteristics, which determines which primary network parameters will be used.
>
>     -   Inference - A single set of primary network weights is used for
>         inference at the chosen rescaling factor. The hypernetwork is no
>         longer needed at this point, and does not contribute to the
>         inference computational cost.
>
>     In our experiments, training a SSHN model takes 10x less time than
>     training the set of Fixed baselines, and is only 1.8x more costly than
>     training a single U-Net model. Once trained, the hypernetwork
>     is only used once to generate the weights of the primary network,
>     which are then used to make the predictions for different inputs, so
>     the computational cost is identical between SSHN and Fixed for a
>     given rescaling factor.
>
> 2.  **Memory Footprint** - We agree that we should say more about memory
>     consumption. Our memory consumption measurements results follow
>     similar trends to the FLOP measurements. Models with smaller
>     rescaling factors substantially reduce memory consumption while
>     maintaining predictive quality for a range of rescaling factors. At
>     training time we sample $\varphi \sim \mathcal{U}(0,0.5)$, so our
>     memory consumption is marginally more than that of a regular U-Net.
>     At inference time, since we do not use the hypernetwork, our memory
>     consumption is no more than that of a regular U-Net.
>
> 3.  **Fixed baseline** - In our experiments, we do not chose a specific
>     rescaling factor for each dataset, as that depends on the downstream
>     accuracy-efficiency considerations. Therefore, in our experiments,
>     we compare methods using the entire accuracy-efficiency frontier.
>     For the Fixed baseline, we evaluate models independently, with each
>     one of them corresponding to a separate rescaling factor and
>     reported an individual datapoint (Figures 3,6 and 7).
>
> 4.  **3D Data** - Yes, in our experiments, we used 2D mid-slices for the
>     3D datasets. We do this because 3D models are more computationally
>     demanding, and the Fixed baseline requires training many individual
>     models with different rescaling factors. Our method can be applied
>     to 3D U-Net models, where the computational and memory improvements
>     might turn out to be even more significant. We believe this is an
>     interesting area of future research.

---

> > ### Comment · Reviewer_wkRF · 2023-08-14
> > **Response by Reviewer wkRF**
> >
> > Dear Authors,
> >
> > thank you for the detailed rebuttal.
> >
> > Here are some of my own comments in line.
> >
> > > Training Cost
> >
> > Noted. My concern was around specifically training time. It seems that training for 1.8x times longer is worth the benefit of being able to produce models on the entire frontier.
> >
> > > 3D Data
> >
> > I believe venturing into 3D might be an ultimate challenge yet desirable for the most important medical applications.
> >
> > I was wondering if you can use your approach for any other hyperparameter, e.g. learning rate?
> >
> > Otherwise, I would like to one more time compliment the authors with a very good paper.

---

> > > ### Author Response · Authors · 2023-08-18
> > >
> > > Thank you for the comments. We believe that there are other efficiency hyperparameters that we could exploit with our amortized learning strategy. For instance, changing the number of downsampling steps or the number of feature channels per layer would change the efficiency characteristics while still preserving a high degree of similarity between the networks. In this work, we focused on the rescaling factor as it presented direct computational benefits, but we are excited about future research expanding these ideas to other hyperparameters.

---

### Official Review · Reviewer_veWP · 2023-07-06

**Soundness:** 4 excellent
**Presentation:** 4 excellent
**Contribution:** 3 good
**Rating:** 7
**Confidence:** 4

**Summary:**

The paper introduces a method that learns a spectrum of CNNs with different rescaling factors. The method relies on using a hyper-network to generate the parameters of the model for a given rescaling factor --- this enables the users to choose the desired accuracy-efficiency trade-off with a single architecture.

While the method is general, the authors demonstrate in two challenging structured prediction tasks (namely, segmentation and registration) the benefits of their approach in terms of accuracy-efficiency trade-off while incurring a small amount of extra training cost.

**Strengths:**


- The proposed method is a creative adoption of hyper network to the practical challenge of enabling the users to choose the desired accuracy-efficiency trade-off at inference time
- Moreover, a comprehensive set of experiments demonstrate SSHN attain considerably better accuracy-efficiency trade-off than relevant baselines in multiple datasets for two challenging tasks
- The manuscript is written very clearly and its motivation is strong.

**Weaknesses:**

- I would like to hear a more detailed explanation as to why learning a single model with varying rescaling factors lead to a consistent improvement in accuracy over the models trained with a fixed rescaling factor.


**Questions:**

- What are the additional computational costs of generating weights through a hyper-network at inference time? Is this included in the calculation of inference costs in Figure7?
- It is unclear why FiLM is a sensible baseline to compare against.
- Is the cost of training a single SSHN smaller than training multiple CNNs independently for fixed rescaling factors? If so, this could be highlighted in the manuscript.
- While the simplicity of the proposed approach is a strength rather than a weakness, one could envision setting different rescaling factors in different parts of the architecture. Such extension seems almost trivial implementationally, but training the hyper-network may become more challenging. It would be great to hear your thoughts on the potential benefits and risks of increasing such degrees of freedoms.

---

> ### Author Rebuttal · Authors · 2023-08-09
>
>
> We appreciate the constructive feedback and comments. Addressing the
> raised questions:
>
> ### Weaknesses:
>
> -   **Accuracy Improvements** - This is an important question, and we
>     don't have a definitive answer. However, the results of the
>     experiment described in Section 5.2, suggest that varying the
>     resolution of intermediate features in the network induces a
>     regularization effect similar to how image scaling techniques in
>     data augmentation pipelines regularize learning.
>
> ### Questions:
>
> -   **Inference Costs** - We would like to clarify how the model
>     development process is performed in our method:
>
>     1.  Training - The hypernetwork is trained by generating the primary
>         network weights from randomly sampled rescaling factors.
>
>     2.  Scale selection - Once trained, the hypernetwork is used once per rescaling factor to predict the weights of each primary network. These weights are used to evaluate the accuracy on a held-out set of data, producing a Pareto accuracy-efficiency frontier.
>     A rescaling factor is then chosen based on the trade-off characteristics, which determines which primary network parameters will be used.
>
>     3.  Inference - A single set of primary network weights is used for
>         inference at the chosen rescaling factor. The hypernetwork is no
>         longer needed at this point, and does not contribute to the
>         inference computational cost.
>
>     Therefore, since the hypernetwork is not used for performing
>     inference, Figure 7 only includes the computational costs of the
>     primary network. We tried to explain this in our method section
>     (L117-119), and will revise our manuscript to better clarify this in
>     the result section as well.
>
> -   **FiLM baseline** - We compare against FiLM because recent work (You
>     only train once: Loss-conditional training of deep networks \[13\])
>     employed FiLM modules as a way to perform amortized learning across
>     multiple loss weightings, efficiently characterizing the loss Pareto
>     frontiers. To the best of our knowledge, this is the closest
>     baseline for the problem statement of learning a family of models in
>     an amortized way.
>
> -   **Training Cost** - Yes, it takes approximately 10x less time to
>     train a single SSHN than to train the set of Fixed baselines
>     independently. We report training times in Table 3 of the
>     supplement. We will revise the text to better highlight this fact.
>
> -   **Different Rescaling Factors** - We agree with the observation. We
>     carried out this experiment and report results for the OASIS dataset
>     in Section C.2 of the supplement. We found that while it is feasible
>     to train a hypernetwork model with separate rescaling factors, it
>     fails to meaningfully improve the accuracy-efficiency Pareto
>     frontier while requiring longer training times to converge.

---

### Official Review · Reviewer_GRpN · 2023-07-08

**Soundness:** 3 good
**Presentation:** 2 fair
**Contribution:** 3 good
**Rating:** 7
**Confidence:** 4

**Summary:**

The paper presents Scale-Space HyperNetworks (SSHN), a method that predicts the weights for a segmentation network for range of rescaling factors. The proposed approach makes it possible to characterize the trade-off between model accuracy and inference efficiency faster, reducing the overall computational cost. Further, the paper demonstrates SSH demonstrate improved generalization on a variety of medical imaging tasks and datasets.

**Strengths:**

- Employing a function h with learnable parameters to map the rescaling ratio to a set of convolutional weights is an interesting and novel approach and the overall framework.
- The paper is evaluated across two critical medical image analysis tasks and is demonstrated to perform better compared to fixed and other variable resizing methods.
- Paper also demonstrate efficiency and other analysis, including varying prior width and weight transferability, which allows deeper understanding of the framework and I'm confident that future works in this direction can benefit from such analysis.



**Weaknesses:**

The weakness that I found in this work is a limited explanation of the method, which might create confusion and readability issues for the readers. Here are a few suggestions/questions to address them:
1. The implementation section should be clearly explained and might need to be expanded to include clear differences between how the network is trained and how it is used in inference or in evaluation. This is important because, in the later part of the paper, there are descriptions (line 198) like "once trained, we use the hyper network to rapidly evaluate a range of phi..".
2. The paper mentions how a hyper-network has more parameters. It would be nice to indicate how much more those parameters.
3. In Fig 2, the hyper network branch goes to both the encoder and decoder. But since there is a concat layer, how can a variable resizing factor be used?
4. The training time result should be included in the main text as this further strengthens the argument of the paper.


**Questions:**

Please look at weakness.

**Limitations:**

Yes.

---

> ### Author Rebuttal · Authors · 2023-08-09
>
>
> We appreciate the constructive feedback and comments. Addressing the
> raised questions:
>
> 1.  **Model Development** - We will revise the manuscript to better
>     explain the model development process and to differentiate between
>     training and inference. To further clarify, the steps we take are:
>
>     -   Training - The hypernetwork is trained by generating the primary
>         network weights from randomly sampled rescaling factors.
>
>     -   Scale selection - Once trained, the hypernetwork is used once per rescaling factor to predict the weights of each primary network. These weights are used to evaluate the accuracy on a held-out set of data, producing a Pareto accuracy-efficiency frontier.
>     A rescaling factor is then chosen based on the trade-off characteristics, which determines which primary network parameters will be used.
>
>     -   Inference - A single set of primary network weights is used for
>         inference at the chosen rescaling factor. The hypernetwork is no
>         longer needed at this point, and does not contribute to the
>         inference computational cost.
>
>     We will revise the text to make this distinction more explicit and
>     clear.
>
> 2.  **Parameter Counts** - We will incorporate additional detail about
>     the parameter counts. In our experiments, the hypernetwork model has
>     approximately 100x more learnable parameters than the Fixed
>     baseline. Importantly, however, the hypernetwork predicts the
>     weights of a primary network that has identical structure as the
>     Fixed baselines.
>
> 3.  **Decoder Resizing** - For resizing layers in the decoding branch,
>     the feature maps are resized to match the spatial dimensions of the
>     tensors from the skip connections that are concatenated afterwards.
>     We will revise the text to better describe this detail of the
>     implementation.
>
> 4.  **Training Runtime** - We will revise the results section to
>     incorporate training runtimes.

---

> > ### Comment · Reviewer_GRpN · 2023-08-20
> > **Official comment**
> >
> > I have read the rebuttal from authors and other reviewers and my score remains unchanged. Thanks!

---

### Decision · Program_Chairs · 2023-09-21

**Decision:**

Accept (poster)

**Comment:**

The paper was reviewed by 5 reviewers. On the one hand, the reviewers appreciated the ideas presented in the paper, found the presentation of the paper clear and highlighted the potential relevance of the network parameter prediction for diverse rescaling factors in medical imaging segmentation.  On the other hand, the reviewers were concerned with the reported experimental results, brought up several relevant prior works and pointed out missing implementation details. Authors submitted the rebuttal. In the post rebuttal phase, four out of five reviewers were inclined towards acceptance. The main criticism of unconvinced reviewer were around paper's novelty and validation. The reviewer brought up relevant prior works. AC agrees with the reviewer that brought up work is indeed relevant. Validation could also be strengthened. The main points of validation criticisms are related to potential underfitting of baselines as the fixed setup is under-performing w.r.t. SSHN. Adding an ablation over the baseline parameters (with parameters growing to the capacity of SSHN) would make the submission stronger. AC concludes that given current experimental setup one of the claims of the paper about the superiority of SSHN ( "... SSHN consistently outperforms regular models by 2-3% for the same rescaling factor.") might not have enough experimental support. Overall, the arguments towards acceptance outweigh the reasons to reject this submission. Thus AC recommends to accept. The AC expects the authors to introduce the reviewers suggestions in the final version of their manuscript. Moreover, the AC asks the authors to (1) either modify the claim in question or include an ablation over parameters to ensure that the baselines are not underfitting to the data and (2) properly credit the related works brought up by the reviewers.